# AI Driven Discovery of Bio Ecological Mediation in Cascading Heatwave Risks

**Yiquan Wang, Qingyun Gao, Yuhan Chang, Jialin Zhang**
Xinjiang University
`{ethan, yun, changyuhan, zhangjialin}@stu.xju.edu.cn`

**Tin-Yeh Huang**
The Hong Kong Polytechnic University
`tin-yeh.huang@connect.polyu.hk`

## Abstract

Compound heatwaves increasingly trigger complex cascading failures that propagate through interconnected physical and human systems, yet the fragmentation of disciplinary knowledge hinders the comprehensive mapping of these systemic risk topologies. This study introduces the Heatwave Discovery Agent HeDA as an autonomous scientific synthesis framework designed to bridge cognitive gaps by constructing a high fidelity knowledge graph from 8,111 academic publications. By structuring 70,297 evidence nodes, the system exhibits enhanced inferential fidelity in capturing long tail risk mechanisms and achieves a significant accuracy margin compared to standard foundation models including GPT 5.2 and Claude Sonnet 4.5 in complex reasoning tasks. The resulting topological analysis reveals a critical bio ecological mediation effect where biological systems function as the primary non linear amplifiers of thermal stress that transform physical meteorological hazards into systemic socioeconomic losses. We further identify latent functional couplings between theoretically distinct sectors such as the heat induced synchronization of power grid failures and emergency medical capacity saturation. These findings elucidate the dynamics of compound climate risks and provide an empirical basis for shifting adaptation strategies from static sectoral defense to dynamic cross system resilience.

## 1 Introduction

Anthropogenic forcing has fundamentally altered the global risk landscape where the aggregation of rainfall extremes and the intensification of oceanic heat potential increasingly converge with thermal anomalies to create compound hazards (Konda et al., 2024; Vissa et al., 2013a;b). Within this evolving thermodynamic regimes heatwaves have emerged as one of the most lethal and socioeconomically devastating hazards facing humanity. The 2003 European heatwave caused over 70,000 deaths and extensive economic losses (Robine et al., 2008; García-Herrera et al., 2010; Beniston, 2004; Black et al., 2004) while the 2021 Pacific Northwest heatwave shattered temperature records and triggered cascading failures across critical infrastructure systems such as power grids and emergency services (White et al., 2023; Saki et al., 2025). Unlike discrete natural disasters heatwaves create complex multidomain cascading effects that propagate through interconnected physical social and economic systems in ways that traditional risk assessment approaches struggle to capture comprehensively (Ebi, 2025). As these compound events become more frequent the inability to foresee how environmental stress transmits to human vulnerability represents a fundamental gap in our collective capacity for climate adaptation.

This challenge is exacerbated by the fragmentation of scientific knowledge where critical risk insights represent long tail knowledge buried within thousands of disconnected studies spanning climatology public health and economics. These specific cascading pathways appear infrequently in general training corpora causing standard large language models to struggle significantly with retrieval and reasoning tasks. Without domain specific fine tuning or external knowledge grounding

even state of the art foundation models often hallucinate or overlook these rare but vital connections (Reichstein et al., 2025). While artificial intelligence offers promising tools for synthesis existing reactive systems lack the ability to effectively capture and reason over this sparse long tail information to identify high impact vulnerabilities that researchers have not yet hypothesized (Ramachandra, 2025; Al Khourdajie, 2025).

We present the Heatwave Discovery Agent HeDA an intelligent multi agent system designed to bridge this gap through automated scientific discovery and multi layer risk propagation analysis. By processing 8,111 academic papers HeDA constructs a comprehensive knowledge graph containing 70,297 nodes and 120,168 relationships (Peng et al., 2023) which serves as a structured repository for capturing long tail scientific evidence. The system exhibits robust reasoning capabilities by achieving 70.0 percent accuracy on complex question answering tasks and showing a substantial performance differential compared to baseline models including GPT 5.2 and Claude Sonnet 4.5 in zero shot settings. This accuracy improvement highlights the efficacy of integrating structured knowledge graphs with large language models to resolve complex domain specific queries without expensive fine tuning thereby facilitating the autonomous discovery of critical risk chains such as the path from atmospheric aerosol accumulation to the overwhelming of tertiary emergency care capacity.

Beyond technical performance our network analysis reveals a critical bio ecological mediation effect where biological systems function as the primary non linear amplifiers of thermal stress transforming physical hazards into systemic economic losses. HeDA further uncovers latent functional couplings between theoretically distinct infrastructure sectors and identifies high impact grey rhino risk chains that represent structurally plausible but historically overlooked transmission vectors. These findings demonstrate the potential of artificial intelligence to decipher the anatomy of cascading climate risks and provide an empirical basis for shifting adaptation strategies from static defense to dynamic system resilience.

## 2 RELATED WORK

### 2.1 COMPOUND WEATHER EXTREMES AND SYSTEMIC RISK PROPAGATION

The characterization of climate hazards has evolved fundamentally from a univariate perspective to a focus on compound weather and climate events where the combination of multiple drivers contributes to societal or environmental risk (Zscheischler et al., 2018; 2020). Unlike isolated meteorological anomalies these concurrent or sequential stressors encompassing aerosol enhanced precipitation events and shifting pollution hotspots (Choudhury et al., 2020; Tyagi et al., 2021) generate non linear cascading effects that propagate through interconnected physical and human systems (Raymond et al., 2020; Niggli et al., 2022). Historical analyses of the 2003 European heatwave and the 2021 Pacific Northwest event indicate that the interaction between atmospheric blocking patterns and surface thermodynamic exchanges involving sensible and latent heat fluxes (Bandaray et al., 2026; Tyagi et al., 2013) creates conditions where thermal stress acts as a systemic multiplier rather than a discrete hazard (Robine et al., 2008; White et al., 2023). Current research underscores that these cascading failures often transcend geographical and sectoral boundaries thereby challenging traditional risk assessment frameworks that treat physical hazards and socioeconomic impacts as linearly related variables (Simpson et al., 2021; Westra & Zscheischler, 2023).

### 2.2 KNOWLEDGE FRAGMENTATION AND THE LIMITS OF DISCIPLINARY MODELING

Despite the growing recognition of systemic risks the capability to model cross-sectoral cascading pathways remains constrained by the fragmentation of disciplinary knowledge. Integrated Assessment Models and standard coupled physical-biogeochemical frameworks frequently struggle to capture the complex adaptive nature of terrestrial ecosystems and their feedback to the climate system (Ackerman et al., 2009; Frank et al., 2015). Studies indicate that single-sector impact models systematically misrepresent the magnitude of risks because they omit the intricate interdependencies between water resources agriculture and energy systems (Harrison et al., 2016; Monier et al., 2018). The requisite knowledge for mapping these complex topologies is distributed across disparate fields ranging from atmospheric physics to epidemiology and public health which creates significant blind spots in identifying the biological and ecological mediators that govern risk transmission (Talukder

et al., 2024; Naylor et al., 2020). This disconnect highlights the critical need for synthesis methodologies capable of identifying latent structural dependencies that are currently obscured by the sheer volume and disciplinary segregation of scientific evidence.

## 2.3 AI-DRIVEN SYNTHESIS AND CAUSAL DISCOVERY IN EARTH SCIENCE

Addressing the challenge of information fragmentation necessitates the deployment of advanced cognitive tools capable of synthesizing vast unstructured scientific corpora into computable formats. The emergence of artificial intelligence for scientific discovery offers a transformative approach to reconstructing these complex risk topologies by moving beyond simple information retrieval to high-dimensional hypothesis generation (Wei et al., 2025; Reddy & Shojaee, 2025). Recent advancements in large language models and structured information extraction facilitate the automated construction of high-fidelity knowledge graphs that can resolve terminological heterogeneity across disciplines and identify causal mechanisms hidden in long-tail literature (Dagdelen et al., 2024; Hartung, 2025). By integrating neuro-symbolic reasoning with domain-specific constraints these autonomous systems provide a novel capability to bridge the gap between physical climate dynamics and socioeconomic vulnerability effectively functioning as a macroscope for detecting emergent systemic risks that human researchers may overlook (Zhu et al., 2024; Li et al., 2025b).

## 3 METHODS

### 3.1 MULTI-AGENT SYSTEM ARCHITECTURE

To overcome the inherent limitations of traditional disciplinary modeling in capturing cross-sectoral cascading risks we developed HeDA as an autonomous cognitive framework designed for Earth system complexity. Unlike standard information retrieval tools which treat scientific evidence as static text strings HeDA functions as a structured synthesis framework that reconstructs the nonlinear topology of heatwave risks by integrating fragmented mechanisms from climatology ecology and economics. The system architecture orchestrates a hierarchical multi-agent workflow where a central coordination unit manages the dynamic scheduling of tasks across extraction construction and reasoning modules as illustrated in Figure 1. This architecture explicitly addresses the challenge of scale mismatch in climate risk assessment by transforming unstructured qualitative descriptions from the abstracts of 8,111 documents into a computable directed graph containing 70,297 nodes. By structuring evidence through this standardized protocol HeDA enables the quantitative analysis of how physical atmospheric anomalies propagate through biological mediators to generate systemic socioeconomic losses.

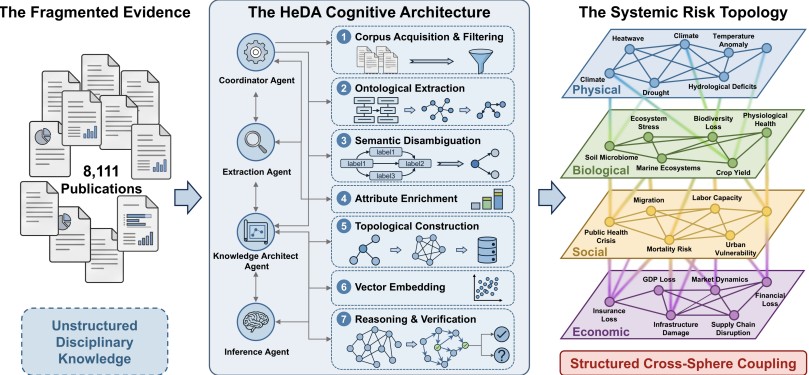

Figure 1: **Architectural workflow of the Heatwave Discovery Agent (HeDA).** The system transforms fragmented unstructured text into a high-fidelity risk topology through a seven-stage neurosymbolic protocol, bridging the gap between physical climate dynamics and socioeconomic impacts.

The synthesis process is governed by a rigorous seven-stage protocol designed to ensure the physical plausibility and semantic consistency of the constructed knowledge graph. The workflow initiates with corpus acquisition and filtering where relevance verification algorithms screen the abstracts to

remove noise and establish a high-fidelity baseline. The subsequent ontological extraction stage utilizes large language models to identify causal triplets while preserving the conditional boundaries of scientific claims. To resolve the terminological heterogeneity inherent in cross-disciplinary research the system executes semantic disambiguation using sentence transformers to cluster synonymous entities such as categorizing thermal stress and heat exposure into unified nodes. The process advances to attribute enrichment where edges are classified into physical social or economic layers followed by topological construction within a graph database environment. The sixth stage involves vector embedding to map discrete entities into a continuous latent space facilitating semantic retrieval. Finally the reasoning and verification stage applies complex graph traversal algorithms to identify latent risk pathways that span multiple systemic layers and validates their structural coherence against established scientific principles.

## 3.2 Multi-layer Risk Propagation Analysis

We establish a mathematical formalism for identifying latent systemic vulnerabilities by treating the risk landscape as a multi layered topology where failures propagate across distinct ontological boundaries. This framework shifts the analytical focus from static node attributes to the dynamic nature of interaction edges effectively capturing the flow of cascading risks through the definition of a mapping function $L$. This function categorizes relationships into four fundamental systemic layers comprising physical climatic processes, biological ecosystem responses, social vulnerability, and economic dependencies.

$$L : E \rightarrow \{\text{Physical}, \text{Biological}, \text{Social}, \text{Economic}, \text{CrossLayer}\}$$

$$\text{where } L(e) = \begin{cases} \text{Physical} & \text{if } e \in \{\text{climatic anomalies, hydrological cycles}\} \\ \text{Biological} & \text{if } e \in \{\text{ecosystem functions, physiological health}\} \\ \text{Social} & \text{if } e \in \{\text{public health, labor capacity, migration}\} \\ \text{Economic} & \text{if } e \in \{\text{infrastructure, market dynamics, GDP}\} \end{cases} \quad (1)$$

While these four layers are ontologically distinct in our knowledge graph construction we recognize that social and economic systems serve as a functionally coupled sink for climate impacts. Consequently in our propagation analysis particularly for evaluating cascading pathways we synthesize the social and economic layers into a unified socioeconomic domain. This hierarchical consolidation allows us to isolate the specific role of biological systems as the critical mediation bridge. To systematically uncover high impact pathways that remain obscured in fragmented literature we define a structural novelty metric for any given propagation chain $P$. This metric operationalizes the concept of the grey rhino risk by synthesizing three theoretical components into a unified score.

$$\text{NoveltyScore}(P) = \alpha \cdot \text{LF}(P) + \beta \cdot \text{CLC}(P) + \gamma \cdot \text{IP}(P) \quad (2)$$

The Literature Frequency component $\text{LF}(P)$ acts as an inverse popularity filter to identify rare co occurrences that represent information theoretic novelty relative to the training corpus as defined by $1 - \frac{f(P)}{F_{\max}}$. The Cross Layer Connectivity component $\text{CLC}(P)$ quantifies the topological complexity of the risk chain by measuring the density of domain transitions utilizing an indicator function $\mathbb{I}$ that rewards pathways traversing multiple system boundaries.

$$\text{CLC}(P) = \frac{\sum_{i=1}^{n-1} \mathbb{I}(L(r_i) \neq L(r_{i+1}))}{n - 1} \quad (3)$$

The Impact Potential $\text{IP}(P)$ integrates the structural centrality of nodes with the confidence levels of extracted relations to ensure that discovered pathways represent physically plausible transmission vectors rather than spurious correlations. We calibrate the weighting parameters $\alpha$, $\beta$, and $\gamma$ to prioritize pathways that exhibit both high structural novelty and significant propagation potential. This mathematical formalism allows the system to detect non linear amplification effects where biological entities serve as the critical transmission bottleneck between physical hazards and socioeconomic stability thereby remapping the search space to highlight latent systemic risks.

### 3.3 RISK DISCOVERY ALGORITHM AND IMPLEMENTATION

Compound heatwaves exert non-linear cascading effects that transcend the resolution boundaries of traditional coupled physical-biogeochemical models where the propagation of vulnerability involves complex feedback loops between atmospheric anomalies and human systems. The interactions between meteorological stressors and socioeconomic losses are rarely direct but are modulated through intricate intermediate ecological buffers that remain computationally opaque in standard sectoral assessments. To resolve this topological blindness we developed a multi-hop risk discovery algorithm that treats the knowledge graph as a high-dimensional state space where cascading failures are modeled as directed propagation paths constrained by specific depth and layer transition parameters. The inference engine implements a parallelized graph traversal strategy that filters out shallow associative correlations with path lengths less than three hops while explicitly prioritizing deep structural dependencies that traverse multiple ontological boundaries from physical hazards through biological systems to economic outcomes. By integrating a structural novelty metric with causality verification, the system isolates high impact low frequency transmission vectors located in the long tail of the probability distribution which characterizes the anatomy of grey rhino risks. This computational framework synthesizes the processed evidence nodes derived from the extensive academic corpus into a unified risk topology that allows for the quantitative identification of the bio ecological mediation layer as the critical bottleneck in systemic failure propagation.

---

**Algorithm 1** Optimized Multi-layer Risk Propagation Discovery

---

**Require:** Knowledge graph $G = (V, E)$, layer mapping $L$, min_hops $d_{\min} = 3$, max_hops $d_{\max} = 5$

**Ensure:** Set of high-novelty risk pathways $\mathcal{P}$

1: Initialize $\mathcal{P} \leftarrow \emptyset$, $visited \leftarrow \emptyset$
2: Select source nodes $S \subset V$ based on high degree centrality
3: **for** each source node $s \in S$ in parallel **do**
4:    $queue.enqueue(\{s\}, 0)$
5:   **while** $queue \neq \emptyset$ **do**
6:     $(path, depth) \leftarrow queue.dequeue()$
7:     **if** $depth < d_{\max}$ **then**
8:       $current \leftarrow path.last()$
9:       **for** each neighbor $n$ of $current$ **do**
10:         $new\_path \leftarrow path.append(n)$
11:         **if** $depth + 1 \geq d_{\min}$ AND CrossLayerCount($new\_path$) $\geq 1$ **then**
12:           $score \leftarrow$ NoveltyScore($new\_path$)
13:           **if** $score > \theta_{\text{novelty}} = 0.7$ **then**
14:             Add $new\_path$ to $\mathcal{P}$
15:           **end if**
16:         **end if**
17:         **if** $new\_path \notin visited$ **then**
18:           $visited.add(new\_path)$
19:           $queue.enqueue(new\_path, depth + 1)$
20:         **end if**
21:       **end for**
22:     **end if**
23:   **end while**
24: **end for**
25: **return** Top-$k$ pathways from $\mathcal{P}$ prioritized by length ($Length \geq 4$) then score

---

### 3.4 DATASET CONSTRUCTION AND TECHNICAL INFRASTRUCTURE

To ensure comprehensive coverage of the heatwave risk landscape we systematically aggregated 8,127 academic records from the Web of Science covering the period from 1968 to 2025. This corpus was subjected to a rigorous quality control protocol where large language models performed relevance verification to filter extraneous data, resulting in a high fidelity dataset of 8,111 processed documents comprising primarily journal articles and book chapters. The resulting knowledge graph

infrastructure is hosted on a Neo4j database environment and powered by Qwen3-Max models for semantic reasoning with FAISS vector indexing, enabling the efficient retrieval and synthesis of 70,297 evidence nodes.

# 4 RESULTS

## 4.1 TOPOLOGICAL RECONSTRUCTION OF COMPOUND RISK LANDSCAPES

The reconstruction of high fidelity risk landscapes necessitates a methodological progression from simple correlation analysis to the identification of causal topologies that transcend disciplinary boundaries. The HeDA framework addresses this challenge by synthesizing fragmented literature into a coherent graph structure that improves inferential accuracy particularly at the critical two hop boundary where physical meteorological anomalies translate into biological impacts as evidenced in Table 1. A rigorous examination of inference depth reveals that the performance differential peaks specifically at this cross disciplinary interface which confirms that the knowledge graph successfully bridges the epistemological silos that traditionally obscure the immediate consequences of meteorological extremes on human systems. The analysis further indicates a sustained performance advantage at three and four hop distances reflecting the system robustness in tracking cascading failures through highly connected mediation nodes such as power grids or ecosystem services. These hub nodes frequently act as information bottlenecks in long chain risk propagation yet the structured reasoning capability allows for the precise mapping of non linear risk accumulation that purely statistical models fail to capture. This topological fidelity establishes the necessary empirical foundation for identifying the latent structural vulnerabilities and bio ecological bottlenecks where the physical dynamics of the Earth system directly intersect with socioeconomic stability.

Table 1: Evaluation of inferential fidelity across varying propagation depths for multi layer risk reconstruction. The stratified accuracy metrics demonstrate the structural advantage of graph integration in resolving complex cascading failure mechanisms that extend beyond immediate physical correlations.

| Model Configuration | KG Aug. | Total Acc (%) | F1-Score | 1-Hop (%) | 2-Hop (%) | 3-Hop (%) | 4-Hop (%) |
|---|---|---|---|---|---|---|---|
| *Standalone Baselines* | | | | | | | |
| GPT-5.2 | No | 47.35 | 0.470 | 64.3 | 29.0 | 48.4 | 47.7 |
| Qwen-Plus | No | 48.85 | 0.487 | 68.2 | 28.6 | 45.6 | 53.0 |
| Claude-Sonnet-4.5 | No | 49.20 | 0.501 | 81.6 | 41.6 | 48.6 | 25.0 |
| Qwen3-Max | No | 50.03 | 0.496 | 67.8 | 35.5 | 47.6 | 49.1 |
| *Knowledge-Augmented* | | | | | | | |
| Qwen-Plus + KG | Yes | 67.42 | 0.676 | 88.3 | 74.5 | 52.7 | 54.2 |
| GPT-5.2 + KG | Yes | 68.20 | 0.683 | 86.4 | 77.7 | 55.9 | 52.8 |
| **HeDA (Qwen3-Max + KG)** | **Yes** | **70.00** | **0.702** | **88.2** | **78.6** | **59.4** | **53.8** |
| Claude-Sonnet-4.5 + KG | Yes | 78.04 | 0.777 | 91.6 | 78.9 | 73.3 | 66.5 |

## 4.2 BIO-ECOLOGICAL AMPLIFICATION MECHANISMS

The non linear nature of compound heatwaves requires a departure from linear impact models to a systemic perspective whereby physical hazards propagate through intermediate layers before manifesting as economic deficits. Topological analysis of the constructed knowledge graph reveals a fundamental structural asymmetry in risk transmission that challenges conventional direct impact assessments by highlighting the role of biological systems as primary stress transducers. Quantitative flux density analysis in Figure 2a demonstrates that while physical systems generate the highest volume of primary triggers with 1137 intra layer associations the subsequent propagation is heavily skewed towards biological targets with 661 transitions compared to only 265 direct physical to economic impacts. This statistical distribution indicates that the magnitude of thermal energy does not directly translate into economic destruction but necessitates a biological substrate to materialize as a systemic loss. The visualization of these cascading flows in Figure 2b further elucidates this mechanism by identifying a critical amplification zone where biological systems specifically agricultural yield and human physiological health function as non linear converters of environmental stress. This bottleneck architecture implies that biological systems absorb the initial shock of meteorological extremes and transduce discrete physical parameters into tangible socioeconomic deficits

through processes such as crop failure and labor capacity reduction. HeDA explicitly identified a latent physiological pathway linking thermal oxidative stress to chronic kidney dysfunction which mirrors the etiology of Chronic Kidney Disease of unknown etiology (CKDu) observed in agricultural regions (Ben Khadda et al., 2024; Sorensen & Garcia-Trabanino, 2019). Recent projections corroborate this AI derived inference by estimating that heat stress induced labor productivity losses could reduce global agricultural output by 18% by the end of the century (Sheng et al., 2025; Casey et al., 2024). Consequently the stability of the economic layer depends less on the absolute intensity of the heatwave than on the resilience thresholds of intermediate ecological mediators implying that current infrastructure centric risk assessments structurally underestimate systemic vulnerability by neglecting these saturation points.

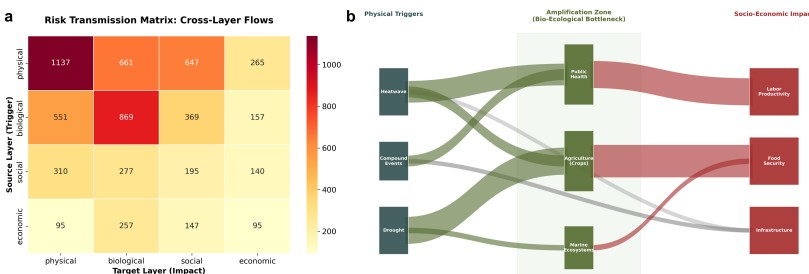

Figure 2: **The Bio-Ecological Mediation Architecture.** (a) The risk transmission matrix quantifies causal density between systemic layers where the high volume of physical-to-biological transitions contrasts sharply with sparse direct physical-to-economic linkages. (b) The Sankey diagram visualizes this topology as a functional bottleneck where biological systems act as the primary amplification zone that transduces meteorological thermal stress into socioeconomic instability.

## 4.3 STRUCTURAL SYNCHRONIZATION OF CROSS-SECTORAL VULNERABILITIES

The systemic risk landscape under extreme thermal regimes exhibits a fundamental topological restructuring whereby theoretically distinct physical and socioeconomic sectors coalesce into integrated vulnerability complexes. Decomposition of these high dimensional interactions utilizing the computational framework reveals in Figure 3a that risk entities do not distribute randomly but self organize into functional clusters that defy standard administrative boundaries. This topological synchronization indicates that independent infrastructure systems such as electrical power grids and emergency medical services effectively collapse into a singular fate community during heatwaves as the thermodynamic failure of cooling mechanisms propagates instantaneously to clinical capacity saturation. A critical examination of the bridge nodes in Figure 3b provides a deeper explanatory mechanism for this coupling by quantifying the brokerage power of specific hazard entities. Notably the analysis identifies marine heatwaves and compound drought events as the dominant coupling agents with the highest betweenness centrality scores rather than purely socioeconomic factors. This structural prominence indicates that compound natural hazards act as the primary topological bridges connecting atmospheric physical anomalies to downstream biological and economic losses. This topological inference was empirically validated by the 2022 compound drought and heatwave event in the Yangtze River Basin where the synchronization of hydrological deficits and heat induced demand surges triggered a 50% reduction in hydropower capacity and consequent industrial shutdowns (Li et al., 2025a; Liu et al., 2023; 2025). Similar spatial compounding effects involving energy linkages have been projected to intensify across major urban clusters under future climate scenarios (Fang et al., 2025; Lv et al., 2024). The high ranking of mortality risk and resilience nodes further corroborates the bio ecological mediation hypothesis suggesting that the stability of the entire risk network functionally depends on the physiological limits of biological organisms and their capacity to withstand compound stressors. These findings imply that the most dangerous propagation vectors reside not within individual sectors but at the latent interfaces where compound environmental extremes force a synchronization between natural ecosystems and human infrastructure.

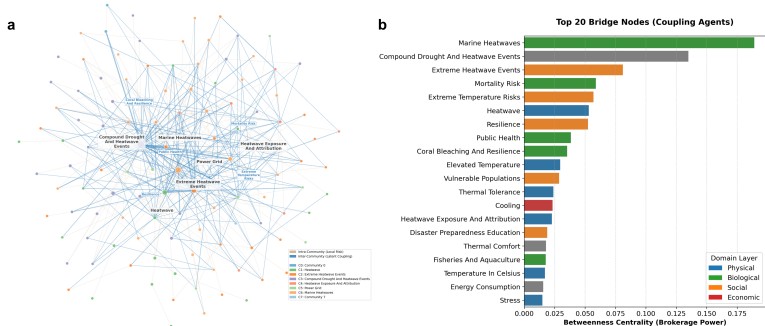

Figure 3: **Topological Anatomy of Cross-Sector Functional Coupling.** (a) The global network structure reveals the emergence of integrated risk communities where distinct physical and social sectors coalesce into synchronized clusters under thermal stress. (b) The ranking of top bridge nodes by betweenness centrality identifies Marine Heatwaves and Compound Drought events as the primary coupling agents that bridge the theoretical gap between physical climate dynamics and socioeconomic impacts.

## 4.4 IDENTIFICATION OF LATENT TRANSMISSION VECTORS

The topological analysis of the constructed knowledge graph facilitates the identification of high impact low frequency transmission vectors that typically evade detection in standard sectoral assessments. These latent cascading pathways represent long tail risks characterized by high structural plausibility within the Earth system network despite occupying a marginal position in existing consensus literature as illustrated in the scatter distribution of Figure 4. Such discovered pathways reveal that systemic vulnerability is not uniformly distributed but disproportionately concentrated within specific bio ecological mediation channels acting as invisible bridges between physical meteorological anomalies and socioeconomic instability. A critical examination of the extracted risk chains elucidates the mechanics of this cross sectoral propagation where biological systems function as non linear amplifiers of thermal stress. In the oceanic domain the computational synthesis identifies a distinct cascading sequence whereby marine heatwaves act as forceful agents of disturbance driving the mass mortality of foundation species such as kelp and seagrass. This AI derived pathway aligns with global observations including the 2014–2016 Northeast Pacific warm anomaly and events in Western Australia where the thermal exceedance of physiological thresholds caused the collapse of habitat forming organisms (Wernberg et al., 2025; Smith et al., 2024; Holbrook et al., 2020). Such ecological degradation fundamentally alters food web structures and erodes essential ecosystem services thereby triggering substantial socioeconomic losses in fisheries and tourism sectors as evidenced by the closure of commercial scallop and crab fisheries following extreme thermal events (Free et al., 2023; Caputi et al., 2019). These findings demonstrate that the thermodynamic energy of oceanic warming modulates through complex intermediate shifts in marine biodiversity rather than translating directly into economic deficits. The system further uncovers obscure physiological feedback loops in terrestrial systems where the impact of extreme temperature is mediated by microscopic biological communities. Recent evidence corroborates the topological finding that heat stress disrupts soil microbiome assembly and suppresses nitrogen fixing bacteria thereby compromising crop resilience and yield independent of water availability (Bei et al., 2023; Elango et al., 2025; Muhammad et al., 2024). This micro scale vulnerability extends to human physiological health where climate induced alterations in gut microbiota composition exacerbate susceptibility to enteric pathogens and metabolic disorders (Litchman, 2025). These findings suggest that dangerous climate risks reside in the blind spots of disciplinary silos where the interplay between physical climatology and biological adaptation creates unexpected failure modes indicating that current adaptation strategies focusing solely on hardening physical infrastructure fundamentally underestimate the fragility of the living systems underpinning economic continuity.

## 5 DISCUSSION & CONCLUSION

The escalating frequency of compound heatwaves necessitates a fundamental reevaluation of risk assessment frameworks as the non linear cascading effects of thermal stress increasingly transcend

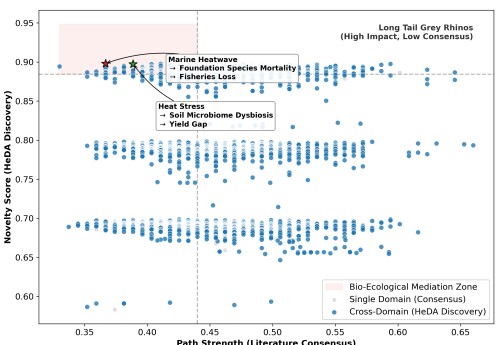

Figure 4: **Identification of Bio-Ecological Grey Rhino Risks in the Long Tail Distribution.** The scatter plot contrasts structural novelty (y-axis) against literature consensus (x-axis), highlighting a distinct Bio-Ecological Mediation Zone (shaded region) in the high-novelty/low-consensus quadrant. Specific annotations identify the two discovered deep-risk pathways discussed in the text: the marine foundation species mortality chain (marked by the red star) and the soil microbiome dysbiosis mechanism (marked by the green star). These outliers validate the capability of HeDA to uncover physically plausible but historically under-researched biological bottlenecks.

the resolution boundaries of traditional coupled physical biogeochemical models. While individual meteorological hazards are well characterized the propagation of these anomalies through interconnected human and natural systems generates systemic failures that are far more destructive than the sum of discrete stressors. Our topological reconstruction of the global risk landscape demonstrates that the interactions between atmospheric dynamics and socioeconomic stability are governed by complex feedback loops that defy linear impact assessments. By synthesizing fragmented mechanisms into a unified high dimensional topology this study reveals that the most critical vulnerabilities reside not within specific sectors but at the latent interfaces where physical climate extremes force a functional synchronization between natural ecosystems and critical infrastructure.

A central finding of this analysis is the identification of a dominant bio ecological mediation effect which fundamentally alters our understanding of how thermodynamic energy translates into systemic loss. We provide empirical evidence that biological systems specifically agricultural productivity and human physiological health function as the primary non linear amplifiers of environmental hazards. The structural topology indicates that physical atmospheric anomalies rarely propagate directly to economic deficits but are instead transduced through the degradation of these biological assets which act as a saturation bottleneck in the global risk network. This mechanism explains why compound events such as heatwaves coinciding with droughts generate disproportionate impacts on food security and labor capacity effectively bridging the theoretical gap between meteorological observation and macroeconomic instability. Consequently adaptation strategies that focus exclusively on hardening physical infrastructure structurally underestimate systemic vulnerability by neglecting the fragility of the living systems that underpin economic continuity.

The topological analysis further elucidates the emergence of synchronized failure modes across theoretically distinct sectors where independent systems coalesce into tightly entangled risk communities under extreme thermal regimes. We observe that compound hazards such as marine heatwaves and hydrological droughts act as coupling agents that dissolve the functional boundaries between energy distribution networks and emergency medical services. This structural interdependence implies that the thermodynamic failure of cooling mechanisms propagates instantaneously to clinical capacity saturation creating a cross domain vulnerability that cannot be mitigated through isolated sectoral planning. These findings highlight the existence of grey rhino risks located in the long tail of the probability distribution which represent structurally plausible but historically overlooked transmission vectors connecting oceanic and atmospheric processes to terrestrial socioeconomic outcomes.

In conclusion this study advances the research on compound weather and climate extremes by establishing a data driven framework for deciphering the anatomy of systemic risk propagation. The utilization of artificial intelligence as an integrative analytical tool enables the synthesis of disparate scientific evidence to uncover the bio ecological bottlenecks and latent functional couplings that

characterize the Anthropocene risk landscape. These insights suggest that effective climate adaptation must evolve from static defense against discrete weather events to a dynamic resilience approach that explicitly strengthens the biological mediators and manages the connectivity between coupled human natural systems. As the interactions between land atmosphere processes and societal feedback loops intensify the ability to anticipate these non linear cascading pathways will be instrumental in securing the stability of global systems against the trajectory of escalating thermal crises.

## FUNDING

This work was supported by the Innovation Practice Program for College Students of the Chinese Academy of Sciences.

## ACKNOWLEDGEMENTS

We would like to express our sincere gratitude to Dr. Yong Ge and Dr. Xilin Wu from the Institute of Geographic Sciences and Natural Resources Research, Chinese Academy of Sciences, for their invaluable guidance and support. This work was supported in part by the Computing and Data Center of Xinjiang University. We acknowledge the computing resources and technical support provided by the Computing and Data Center of Xinjiang University.

## DATA AVAILABILITY

The code used in this article is available in the GitHub repository: `https://github.com/wyqmath/heatwave`.

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
