# OpenReview forum: "AI Driven Discovery of Bio Ecological Mediation in Cascading Heatwave Risks"
_ICLR.cc/2026/Workshop/FM4Science — ICLR 2026 Workshop FM4Science Poster_

### Official Review · Reviewer_fxj7 · 2026-02-15
**Technically Substantial with Good Originality**

**Rating:** 8
**Confidence:** 3

**Review:**

This paper introduces HeDA (Heatwave Discovery Agent), an AI system that extracts and aggregates causal relationships from 8,111 climate-related scientific papers into a large-scale knowledge graph (≈70K nodes, ≈120K edges) to identify cascading multi-layer heatwave risks. The framework assigns system-layer labels (Physical, Biological, Social, Economic) and computes a “NoveltyScore” to detect underexplored but high-impact cross-layer pathways. The central scientific claim is that biological systems serve as mediators amplifying heatwave impacts into socio-economic consequences.

---

### Official Review · Reviewer_uHj7 · 2026-02-18
**AI-Driven Discovery of Bio-Ecological Mediation in Cascading Heatwave Risks**

**Rating:** 6
**Confidence:** 3

**Review:**

This paper introduces HeDA, a multi-agent cognitive framework that constructs a high-fidelity knowledge graph from scientific literature to uncover the non-linear role of biological systems in mediating cascading heatwave risks.

**Pros:**
- The integration of a multi-agent system with domain-specific knowledge graphs effectively bridges the fragmented gap between physical climate dynamics and socioeconomic impact assessments.
- The proposed "NoveltyScore" metric innovatively combines information theory and topological connectivity to successfully identify high-impact, low-frequency "grey rhino" risks.
- The topological analysis offers a significant theoretical contribution by empirically demonstrating that biological systems function as the primary non-linear amplifiers transforming thermal stress into systemic economic losses.
- The system demonstrates superior inferential fidelity in complex reasoning tasks compared to standard foundation models by utilizing structured evidence nodes to reduce hallucinations.

**Cons:**
- The corpus size of 8,111 publications appears relatively limited for a comprehensive global synthesis of cross-disciplinary literature, potentially missing obscure but critical risk pathways.
- The evaluation methodology relies heavily on accuracy metrics against other LLMs, lacking a robust human-expert validation phase to rigorously verify the causality of the discovered latent connections.
- The technical description of the multi-agent coordination and specific prompt engineering strategies for the "Reasoning & Verification" stage is insufficient for full reproducibility.
- The generalization capability of the proposed framework to other compound climate hazards beyond heatwaves remains theoretically discussed but experimentally unverified.

---

### Official Review · Reviewer_r65y · 2026-02-19
**LLM-Augmented Knowledge Graph for Heatwave Risk Synthesis — Interesting Application but Undermined by Evaluation Circularity and Overstated Claims**

**Rating:** 5
**Confidence:** 3

**Review:**

### Summary

This paper introduces HeDA (Heatwave Discovery Agent), a multi-agent system that constructs a knowledge graph (70,297 nodes, 120,168 relationships) from 8,111 heatwave-related academic abstracts and performs graph traversal to identify cross-sectoral cascading risk pathways. The system uses Qwen3-Max for triplet extraction and semantic reasoning, Neo4j for graph storage, and FAISS for vector retrieval. The main scientific claim is the identification of a "bio-ecological mediation effect" — that biological systems (agriculture, human physiology) act as non-linear amplifiers that mediate the propagation of physical heatwave hazards into socioeconomic losses, rather than direct physical-to-economic transmission. The system is evaluated on a custom QA benchmark stratified by reasoning depth (1–4 hops), where knowledge-augmented models outperform standalone LLMs. The paper also presents network topology analysis (cross-layer flow matrices, betweenness centrality, community structure) and identifies specific risk chains (marine heatwave → foundation species mortality → fisheries collapse; heat stress → soil microbiome disruption → crop failure).

### Pros

**1. Timely and important problem framing.** The challenge of synthesizing fragmented, cross-disciplinary climate risk knowledge is real and well-motivated. The paper correctly identifies that cascading heatwave impacts traverse physical, biological, and socioeconomic domains in ways that siloed research struggles to capture. The references to the 2003 European heatwave, 2021 PNW event, and 2022 Yangtze drought-heat compound event are appropriate and well-chosen.

**2. Reasonable system architecture.** The seven-stage pipeline (corpus filtering → ontological extraction → semantic disambiguation → attribute enrichment → topological construction → vector embedding → reasoning) is a sensible workflow for KG construction from scientific abstracts. The use of sentence transformers for entity disambiguation and the four-layer ontological classification (Physical/Biological/Social/Economic) provide useful structure.

**3. The cross-layer flow analysis yields intuitive results.** Figure 2's risk transmission matrix — showing 661 physical→biological transitions vs. only 265 physical→economic direct links — provides a clear quantitative summary of the mediation hypothesis. The Sankey diagram effectively visualizes the bottleneck architecture. Whether this constitutes a "discovery" is debatable (see Cons), but as a structured literature synthesis it has value.

**4. Useful mathematical formalization.** The NoveltyScore metric (Eq. 2) combining literature frequency, cross-layer connectivity, and impact potential is a reasonable heuristic for identifying under-studied but structurally plausible risk chains. The multi-hop traversal algorithm (Algorithm 1) with minimum depth and cross-layer constraints is clearly specified.

### Cons

**1. HeDA is outperformed by its own baseline in Table 1.** This is the most critical issue. Claude-Sonnet-4.5 + KG achieves 78.04% total accuracy vs. HeDA (Qwen3-Max + KG) at 70.00% — a gap of 8 percentage points. Yet the abstract and introduction frame HeDA as the main system contribution, claiming "a significant accuracy margin compared to standard foundation models." This claim is technically only true for the *standalone* (no-KG) comparisons. The actual lesson from Table 1 is that **the knowledge graph is the contribution, not the agent architecture** — any LLM benefits substantially from KG augmentation, and a stronger base LLM (Claude) with the same KG outperforms HeDA by a wide margin. The paper does not acknowledge or discuss this inconvenient result, which seriously undermines the credibility of the presentation.

**2. Circular evaluation design.** The QA benchmark used to evaluate inferential fidelity (Table 1) is constructed from the same knowledge graph that HeDA uses for augmentation. The paper does not describe how these questions were generated, who validated them, or whether they test knowledge beyond what the KG explicitly encodes. This creates a fundamental circularity: KG-augmented models perform better on questions derived from the KG, which is expected by construction and does not demonstrate genuine reasoning capability. Without an independently constructed, expert-validated benchmark, the accuracy numbers are uninformative about real-world utility.

**3. The "discoveries" are well-established in existing literature.** The two highlighted risk chains — (a) marine heatwaves → foundation species mortality → fisheries/ecosystem collapse and (b) heat stress → soil microbiome disruption → crop failure — are extensively documented in the very references the paper cites (Wernberg et al. 2025; Smith et al. 2024; Free et al. 2023; Bei et al. 2023). Similarly, the link between heat exposure and CKDu in agricultural workers is the subject of a NEJM perspective (Sorensen & Garcia-Trabanino 2019) cited in the paper itself. Calling these "AI-driven discoveries" or "latent transmission vectors" significantly overstates the novelty. At best, HeDA rediscovered well-known pathways from its input corpus; the paper provides no evidence that it identified genuinely novel connections absent from the literature.

**4. No validation of knowledge graph quality.** The paper reports aggregate statistics (70,297 nodes, 120,168 relationships) but provides no evaluation of extraction quality — no precision/recall on triplet extraction, no human evaluation of edge accuracy, no inter-annotator agreement, no comparison to a manually curated gold standard. For a system whose entire downstream analysis depends on KG fidelity, this is a major omission. The seven-stage pipeline sounds rigorous in description, but without quantitative quality metrics, the reader has no way to assess whether the graph contains systematic biases, hallucinated edges, or missing relationships.

**5. Key parameters and design choices are unspecified.** The weighting parameters α, β, γ in the NoveltyScore (Eq. 2) are described as "calibrated" but their values are never reported. The novelty threshold θ = 0.7 appears without justification. The Impact Potential component IP(P) is described only vaguely ("integrates structural centrality with confidence levels"). The number of source nodes |S| selected for traversal, the value of k for top-k pathway selection, and the specifics of the "causality verification" step are all missing. This makes the work unreproducible.

**6. The "bio-ecological mediation" finding may be an artifact of corpus composition.** The paper constructs its corpus from heatwave-related literature, which by definition emphasizes the biological and health impacts of heat. If the input literature disproportionately studies biological pathways (because that's what heatwave researchers study), then finding that biological nodes dominate the KG topology is a reflection of publication patterns, not an objective property of risk propagation. The paper does not control for this sampling bias or consider whether the observed flow asymmetry (physical→biological >> physical→economic) simply reflects the disciplinary composition of the input corpus.

**7. Relationship to prior preprint raises novelty concerns.** The system name "HeDA," the multi-agent architecture, the multi-layer risk propagation formalism, and the core methodology closely mirror a preprint (arXiv:2509.25112, September 2025) that describes HeDA processing 10,247 papers to construct a KG with 23,156 nodes and achieving 78.9% accuracy. The present submission uses different numbers (8,111 papers, 70,297 nodes, 70.0% accuracy) but the conceptual framework appears substantially similar. The relationship between these two works is not discussed, and the differing statistics are unexplained.

**8. Poor workshop fit for FM4Science.** The FM4Science workshop focuses on foundation models for science — physical inductive biases, domain-specific architectures, multi-modal scientific FMs, uncertainty quantification. HeDA is a RAG/KG-augmented literature synthesis pipeline that uses an off-the-shelf LLM (Qwen3-Max) as a black-box text processor. There is no foundation model being developed, no scientific prior being embedded in model architecture, and no uncertainty quantification over the extracted knowledge or discovered pathways. The contribution is closer to "NLP for systematic review" than "foundation models for science."

**9. Writing is excessively verbose and repetitive.** The paper restates the same core claim (biological systems mediate heatwave risk propagation) in nearly identical language across the abstract, introduction, Section 4.2, Section 4.3, Section 4.4, and Section 5. Phrases like "non-linear amplifiers of thermal stress," "latent functional couplings," and "grey rhino risks" appear repeatedly without adding new information. Significant space is devoted to rhetorical framing rather than technical detail. This makes the paper feel padded relative to its actual methodological content, and the missing technical details (see Con 5) suffer as a result.

### Minor Issues
- The paper processes only abstracts, not full texts. This is acknowledged implicitly ("from the abstracts of 8,111 documents") but its implications for KG completeness are not discussed — abstracts often omit mechanistic details and quantitative findings that would enrich risk pathway characterization.
- Several references appear to be self-citations that could compromise anonymity (Bandaray et al., 2026; Tyagi et al., 2013; Vissa et al., 2013a;b; Konda et al., 2024; Choudhury et al., 2020; Tyagi et al., 2021 — many sharing co-authors).
- The 4-hop accuracy for standalone Claude (25.0%) vs. standalone GPT-5.2 (47.7%) in Table 1 seems anomalous and is unexplained.
- Algorithm 1 uses BFS without any priority mechanism, yet the return statement prioritizes by length then score — the interaction between exhaustive BFS and this filtering is unclear.

---

### Decision · Program_Chairs · 2026-03-02

Accept (Poster)